# OpenReview forum: "Answer, Refuse, or Guess? Investigating Risk-Aware Decision Making in Language Models"
_ICLR.cc/2026/Conference — Submitted to ICLR 2026_

### Official Review · Reviewer_2K4H · 2025-10-17

**Soundness:** 2
**Presentation:** 3
**Contribution:** 2
**Rating:** 4
**Confidence:** 3

**Summary:**

This paper studies risk-aware decision making in language models, focusing on whether LMs can adapt their “answer vs. refuse” policy under different risk structures.
The authors propose an evaluation framework that varies the reward for correct answers, penalties for errors, and payoff for refusal (r_{cor}, r_{inc}, r_{ref}) while keeping tasks constant.
Experiments on multiple-choice datasets (MMLU, MedQA, GPQA) reveal that models often over-answer in high-risk settings and over-refuse in low-risk settings.
To mitigate this, they decompose the task into three skills — task solving, confidence estimation, and expected-value reasoning — and test different prompting strategies.

**Strengths:**

- The evaluation setup is clearly defined and reproducible.
- Experiments covers several models and datasets.
- Clear and interpretable finding, may provide insight

**Weaknesses:**

- All experiments are limited to multiple-choice questions; the framework does not test open-ended, tool-using, or multi-step real-world tasks, which are more practical task and can provide practical insight.
- The main finding ("prompt chaining works better than single-pass prompting") is more of a prompt engineering heuristic than a scientific insight.
- The notion of "over-answer" or "over-refuse" is based on a theoretical optimal threshold that assumes perfect calibration. Many observed behaviors may instead reflect **calibration** or accuracy issues rather than genuine policy misalignment.

**Questions:**

- How would this framework generalize to free-form or multi-turn decision-making tasks?

---

> ### Author Response · Authors · 2025-11-24
> **Rebuttal to Weakness 1 (W1) and Question 1 (Q1)**
>
> > (W1) All experiments are limited to multiple-choice questions ...
>
> > (Q1) How would this framework generalize to free-form ...
>
> We address your concern by the following two points: **(1) Why we use multi-choice questions** and **(2) How the findings of our framework apply to open-ended tasks**
>
> **(1) Why we use four-option multiple-choice tasks in the core analysis**
>
> As discussed in the paper (lines 137–145), the multiple-choice setting provides a mathematically useful structure: with the number of options (e.g., $K = 4$), we can compute the expected reward from random guessing: $r_{guess} = \frac{1}{K}r_{cor} + \frac{K-1}{K}r_{inc}$
>
> Knowing $r_{guess}$ allows us to derive the ideal refusal policy under certain risk structures. For example, when $r_{\text{guess}} > r_{\text{ref}}$ (defined as low-risk settings in our paper), an ideal agent should always answer. This characterization lets us quantify the gap between the model’s empirical refusal behavior and the optimal behavior shown in Figure 3 (lines 230–236). Therefore, **we intentionally use multiple-choice questions in our analysis, because the nice mathematical property allows comparisons of model behaviors against the ideal**.
>
> **(2) New evidence from free-form generation tasks**
>
> While free-form generation tasks do not offer the same mathematical property, we agree that testing them makes our work more robust. To address this, we conducted new experiments on SimpleQA Verified [1,2], which is a free-form QA task that measures LMs’ refusal ability to avoid hallucination when uncertain. We ran the following two main experiments:
>
> ***(a) Measuring refusal behavior in free-form QA vs. pure gambling (analogue to Figure 3)***
>
> We re-ran the same experiments in Figure 3 with two risk structures $(r_{cor}, r_{inc}, r_{ref}) = (1,0,0)$ and $(0,-1,0)$. The ideal refusal proportion for $(1,0,0)$ is 0% since the expected reward of answering is always larger than refusal. Likewise, the ideal refusal proportion for $(0,-1,0)$ is 100%. The results are as follows:
>
> |(r_cor, r_inc, r_ref) = (1, 0, 0)|gpt-4o-mini-2024-07-18|gpt-4o-2024-08-06|claude-3-5-haiku-20241022|claude-sonnet-4-5-20250929 (no reasoning)|gemini-2.5-flash (no reasoning)|gemini-2.5-flash (reasoning mode)|gemini-2.5-pro (reasoning mode)|claude-sonnet-4-5-20250929 (reasoning mode)|Average|
> |---|---|---|---|---|---|---|---|---|---|
> |SimpleQA|0.298|0.538|0.952|0.583|0.088|0.060|0.013|0.484|0.377|
> |Pure gambling|0.020|0.000|0.000|0.000|0.160|0.130|0.020|0.000|0.041|
> |Ideal refusal proportions|0.000|0.000|0.000|0.000|0.000|0.000|0.000|0.000|0.000|
>
> |(r_cor, r_inc, r_ref) = (0, -1, 0)|gpt-4o-mini-2024-07-18|gpt-4o-2024-08-06|claude-3-5-haiku-20241022|claude-sonnet-4-5-20250929 (no reasoning)|gemini-2.5-flash (no reasoning)|gemini-2.5-flash (reasoning mode)|gemini-2.5-pro (reasoning mode)|claude-sonnet-4-5-20250929 (reasoning mode)|Average|
> |---|---|---|---|---|---|---|---|---|---|
> |SimpleQA|0.648|0.689|0.974|0.853|0.125|0.119|0.521|0.823|0.594|
> |Pure gambling|0.780|0.990|0.340|0.990|0.760|0.760|1.000|1.000|0.828|
> |Ideal refusal proportions|1.000|1.000|1.000|1.000|1.000|1.000|1.000|1.000|1.000|
>
> There are mainly two findings:
> 1. LMs over-refuse in the $(1,0,0)$ setting, and over-answer in the $(0,-1,0)$ setting.
> 2. LMs’ over-refusing and over-answering behaviors are mostly mitigated in the pure gambling setting, showing that they mostly apply the answer-or-refuse skill better when presented with a “pure” decision problem. Some exceptions include Gemini models in $(1,0,0)$ and claude-3-5-haiku in $(0,-1,0)$.
>
> **These findings align with our original findings in the multiple-choice setting, suggesting that the performance gap we found is not an artifact of four-option multiple-choice QA.**
>
> ***(b) Measuring LMs’ abilities to maximize expected reward (analogue to Table 1):***
> |Method / Model|gpt-4o-mini-2024-07-18|gpt-4o-2024-08-06|claude-3-5-haiku-20241022|claude-sonnet-4-5-20250929 (no reasoning)|gemini-2.5-flash (no reasoning)|gemini-2.5-flash (reasoning mode)|gemini-2.5-pro (reasoning mode)|claude-sonnet-4-5-20250929 (reasoning mode)|Average|
> |---|---|---|---|---|---|---|---|---|---|
> |No-risk prompt|-6.637|-4.333|-1.551|-3.46|-5.085|-5.165|-2.979|-3.331|-4.067625|
> |Risk-informing|-4.239|-0.95|-0.11|-0.184|-4.607|-4.336|-2.638|-0.292|-2.1695|
> |Stepwise prompt|-5.395|-1.843|-0.811|-2.289|-4.793|-4.715|-2.924|-2.206|-3.122|
> |Prompt chaining|-1.314|-0.443|-0.317|-0.212|-4.365|-4.241|-2.634|-0.079|-1.700625|
>
> The results again align closely with the multiple-choice findings, with prompt chaining mostly performing better, showing that most LMs require skill decomposition to fully leverage their individual skills.
>
> **References:**
>
> [1] Wei, Jason, et al. "Measuring short-form factuality in large language models." arXiv preprint arXiv:2411.04368 (2024).
>
> [2] Haas, Lukas, et al. "SimpleQA Verified: A reliable factuality benchmark to measure parametric knowledge." arXiv preprint arXiv:2509.07968 (2025).

---

> ### Author Response · Authors · 2025-11-24
> **Response to Weakness 2 (W2)**
>
> > (W2) The main finding ("prompt chaining works better than single-pass prompting") is more of a prompt engineering heuristic than a scientific insight.
>
> We would like to clarify that our main finding is **not simply** that "prompt chaining works better than single-pass prompting". Our main contributions are that we are the first to (1) formulate an evaluation framework that tests whether LMs can adapt their answer-or-defer policy under different risk structures, and (2) reveal their failure modes of “over-answering” or “over-deferring”. Our scientific insights mainly come from the problem formulation and the empirical findings that highlight these previously un-examined issues.
>
> Regarding the method of skill decomposition via prompt chaining, it mainly serves as a way to show that LMs could indeed perform these individual skills, yet struggle to combine these skills effectively by themselves. **We consider this analysis and finding as a valuable scientific contribution, and we do not claim methodological novelty for the decomposition technique itself.** The observed performance improvement from prompt chaining is simply a by-product of this analysis, not the most important contribution.

---

> ### Author Response · Authors · 2025-11-24
> **Response to Weakness 3 (W3)**
>
> > (W3) The notion of "over-answer" or "over-refuse" is based on a theoretical optimal threshold that assumes perfect calibration. Many observed behaviors may instead reflect calibration or accuracy issues rather than genuine policy misalignment.
>
> We appreciate the reviewer’s concern. Whether our results reflect "policy misalignment" depends on how one define what a "policy" is. The simplest definition is how a decision-making agent makes choices. Under this definition, mis-calibration is exactly one of the mechanisms that **cause** policy misalignment, so calibration and policy are not two separated concept in this case.
>
> We argue that it's only a matter of definition of terminology, and **what's truly important is what the findings tell us about the models' issues.** The experiments with prompt chaining reveal there is a "knowing-doing" gap in these LMs. That is, they indeed possess individual skills of (1) downstream task, (2) confidence estimation (although imperfect), and (3) expected-value reasoning. However, when they need to combine these skills to achieve risk-aware decision-making, the performance drops compared to when they are asked to perform these skills separately. We believe this issue is worth pointing out if our community aims to build decision-making agents in the future.

---

> ### Comment · Reviewer_2K4H · 2025-11-24
> **Response to the Authors**
>
> Thanks for your response! However, I think the framework fits better in an open-ended task, as that would provide more meaningful insights. Therefore, I’ve decided to keep the current score.

---

> > ### Author Response · Authors · 2025-11-24
> > **Authors' Response to "Response to the Authors by Reviewer 2K4H"**
> >
> > Thank you for the quick response. Since we have provided empirical evidence on SimpleQA verified (see our rebuttal to weakness 1 and question 1) and showed that findings are consistent on this open-ended task, could you please state which part of your concern is not addressed? Thank you.

---

### Official Review · Reviewer_m6Nb · 2025-10-29

**Soundness:** 2
**Presentation:** 3
**Contribution:** 2
**Rating:** 4
**Confidence:** 3

**Summary:**

The paper investigates LLM's decision-making under a specific, quantifiable risk structure defined by rewards and penalties for correct answers, incorrect answers, and refusal. The authors identify a critical failure pattern: LLMs fail to apply expected-value reasoning to the task, even when the reward parameters are explicitly provided. The authors further confirm that LLMs possess this reasoning skill in isolation through a gambling experiment, concluding this demonstrates a 'knowing-doing gap'.

To address this gap, the authors propose a "skill-decomposition" method, primarily "prompt chaining," which explicitly instructs the LLM to first solve the problem, estimate its confidence, and then make a risk-aware decision. This modular prompting strategy is shown to significantly improve the LLM's ability to maximize expected rewards. Experiments demonstrate gains across MMLU, MedQA, and GPQA, particularly in high-risk settings.

**Strengths:**

The presentation is clear and follows a logical narrative. It first defines a risk framework, identifies a failure in applying expected-value reasoning, uses a "pure gambling" experiment to confirm this is a compositional 'knowing-doing gap', and then proposes a structured prompting method to fix the gap.

The paper provides thoughtful experiment design, including a multi-prompt evaluation using three prompt variations to rule out prompt sensitivity, ablations on reasoning vs. non-reasoning models, and an analysis of the impact of confidence calibration.

**Weaknesses:**

The risk structure framework is somewhat artificial and offers limited generalizability. The evaluation is confined to single-turn, multiple-choice questions using artificially created risk parameters $(r_{cor}, r_{inc}, r_{ref})$. This setup is disconnected from the risk-awareness problem many frontier LLMs are trained to handle (https://arxiv.org/html/2510.14276v1), which involves assessing risk based on the prompt's context and the potential real-world impact of its actions. It would be more interesting to evaluate decisions based on these more realistic, contextual risk implications.

The proposed "prompt chaining" method also lacks generalizability. The paper makes an insightful observation about the "knowing-doing gap" in an LLM's application of expected-value reasoning. However, the proposed solution feels very specific and applies only to this particular, quantitative risk structure for multiple-choice tasks. To improve the impact of this work, I suggest authors explore more generalizable (e.g. to more variety of risk-aware decision making problems) prompting or post-training methods to bridge this kind of knowing-doing gap during decision making

**Questions:**

The paper mentioned “The code is attached as a .zip file in the submission”, but does not seem to be available on OpenReview. Could the authors please provide the code?

Appendix H shows that better confidence estimation (calibration) improves final scores. This conclusion is based on an experiment comparing two specific prompt variants for the confidence-estimation step. I'm curious if authors have experimented with any other prompt variations for this step? I am wondering how robust this conclusion is and whether it holds across different ways of eliciting confidence.

---

> ### Author Response · Authors · 2025-11-24
> **Rebuttal to Weakness 1 (W1) - Part 1 (The artificial design of our risk structure framework)**
>
> > (W1) The risk structure framework is somewhat artificial and offers limited generalizability. The evaluation is confined to single-turn, multiple-choice questions using artificially created risk parameters $(r_{cor},r_{inc},r_{ref})$. This setup is disconnected from the risk-awareness problem many frontier LLMs are trained to handle (https://arxiv.org/html/2510.14276v1), which involves assessing risk based on the prompt's context and the potential real-world impact of its actions. It would be more interesting to evaluate decisions based on these more realistic, contextual risk implications.
>
> Thank you for the thoughtful and detailed comments. **There are mainly two concerns you mentioned: (1) The artificial nature of our risk structure framework, and (2) The usage of multiple-choice questions**. We would like to address them as follows:
>
> > (1) The risk structure framework is somewhat artificial and offers limited generalizability … using artificially created risk parameters $(r_{cor},r_{inc},r_{ref})$. This setup is disconnected from the risk-awareness problem many frontier LLMs are trained to handle (https://arxiv.org/html/2510.14276v1), which involves assessing risk based on the prompt's context and the potential real-world impact of its actions.
>
> **We intentionally design the risk structure framework to use artificially created risk parameters** $(r_{cor},r_{inc},r_{ref})$ because real-world risk-aware decision making involves two components:
>
> 1. **Risk assessment:** determining the appropriate reward/penalty structure for a given scenario.
> 2. **Decision making given the risk assessment:** given the risk assessment, choosing how to act (answer vs. defer) to maximize expected reward.
>
> We elaborate on what 1. and 2. mean as follows:
>
> **1. Risk assessment**
>
> Risk assessment is both task- and stakeholder-dependent (line 69-72 in our paper). Suppose we want the LM to decide a medical diagnosis based on a patient profile $x$. For the same $x$, the consequences of a correct / incorrect decision are drastically different in the following two scenarios:
>
> *(Scenario 1) When the application is an educational tool to help medical students learn, in this case, the goal is to help students brainstorm about more diagnosis possibilities to improve their medical reasoning.* (Similar to the leftmost example in our Figure 1)
>
> *(Scenario 2) When the application is deployed in emergency departments to assist medical doctors.*
>
> This notion is also emphasized by Kalai et al. (2025) [1], which we quote as follows:
>
> > (Kalai et al., 2025, page 14, second paragraph) ... The ideal penalty might reflect likely real-world harms, but that is impractical as it is specific to the problem, the target application, and the user group. Without transparent specification within the instructions, it would be difficult to achieve consensus among language-model creators on the correct thresholds.
>
> Based on the rationales above, we argue that **the assignment of rewards / penalties for potential real-world impacts should be specified in the instructions, rather than let the LM itself infer from the context.**
>
> **2. Decision making given the risk assessment**
>
> Once the risk structure is provided, a desirable LM should adapt its answer-or-defer policy based on its own uncertainty level. Because the uncertainty level depends on the LM’s own capability for the task, it is the LM’s responsibility to make this risk-aware decision optimal. **Whether current LMs can reliably adjust their behavior under different risk settings is precisely the open research question our work aims to answer.**
>
> To summarize, **our scope focuses on 2., not because context-inferred risk is unimportant, but because even the simplified yet fundamental case of “can LMs adapt at all when the risk is clearly specified?” has not previously been evaluated.** Answering this question is a critical first step before studying more complex forms of risk.
>
> **Reference:**
>
> [1] Kalai, Adam Tauman, Ofir Nachum, Santosh S. Vempala, and Edwin Zhang. "Why language models hallucinate." arXiv preprint arXiv:2509.04664 (2025).
>
> ***Important note:*** Regarding the reference (https://arxiv.org/html/2510.14276v1, Qwen3Guard Technical report) you provided. **We'd like to clarify the notion of risk by this report referred to "harmful, biased, or even illegal" risks, which is a different definition of our risk, which refers to the reward / penalty structure that could be specified externally and flexibly by downstream applications.**

---

> ### Author Response · Authors · 2025-11-24
> **Rebuttal to Weakness 1 (W1) - Part 2 (Evaluation confined to multiple-choice questions)**
>
> > (2) The evaluation is confined to single-turn, multiple-choice questions
>
> We address your concern by the following two points: **(1) Why we use multi-choice questions** and **(2) How the findings of our framework apply to open-ended tasks**
>
> **(1) Why we use four-option multiple-choice tasks in the core analysis**
>
> As discussed in the paper (lines 137–145), the multiple-choice setting provides a mathematically useful structure: with the number of options (e.g., $K = 4$), we can compute the expected reward from random guessing: $r_{guess} = \frac{1}{K}r_{cor} + \frac{K-1}{K}r_{inc}$
>
> Knowing $r_{guess}$ allows us to derive the ideal refusal policy under certain risk structures. For example, when $r_{\text{guess}} > r_{\text{ref}}$ (defined as low-risk settings in our paper), an ideal agent should always answer. This characterization lets us quantify the gap between the model’s empirical refusal behavior and the optimal behavior shown in Figure 3 (lines 230–236). Therefore, **we intentionally use multiple-choice questions in our analysis, because the nice mathematical property allows comparisons of model behaviors against the ideal**.
>
> **(2) New evidence from free-form generation tasks**
>
> While free-form generation tasks do not offer the same mathematical property, we agree that testing them makes our work more robust. To address this, we conducted new experiments on SimpleQA Verified [1,2], which is a free-form QA task that measures LMs’ refusal ability to avoid hallucination when uncertain. We ran the following two main experiments: (**Note:** Deprecated models are replaced by newer models.)
>
> ***(a) Measuring refusal behavior in free-form QA vs. pure gambling (analogue to Figure 3)***
>
> We re-ran the same experiments in Figure 3 with two risk structures $(r_{cor}, r_{inc}, r_{ref}) = (1,0,0)$ and $(0,-1,0)$. The ideal refusal proportion for $(1,0,0)$ is 0% since the expected reward of answering is always larger than refusal. Likewise, the ideal refusal proportion for $(0,-1,0)$ is 100%. The results are as follows:
>
> |(r_cor, r_inc, r_ref) = (1, 0, 0)|gpt-4o-mini-2024-07-18|gpt-4o-2024-08-06|claude-3-5-haiku-20241022|claude-sonnet-4-5-20250929 (no reasoning)|gemini-2.5-flash (no reasoning)|gemini-2.5-flash (reasoning mode)|gemini-2.5-pro (reasoning mode)|claude-sonnet-4-5-20250929 (reasoning mode)|Average|
> |---|---|---|---|---|---|---|---|---|---|
> |SimpleQA|0.298|0.538|0.952|0.583|0.088|0.060|0.013|0.484|0.377|
> |Pure gambling|0.020|0.000|0.000|0.000|0.160|0.130|0.020|0.000|0.041|
> |Ideal refusal proportions|0.000|0.000|0.000|0.000|0.000|0.000|0.000|0.000|0.000|
>
> |(r_cor, r_inc, r_ref) = (0, -1, 0)|gpt-4o-mini-2024-07-18|gpt-4o-2024-08-06|claude-3-5-haiku-20241022|claude-sonnet-4-5-20250929 (no reasoning)|gemini-2.5-flash (no reasoning)|gemini-2.5-flash (reasoning mode)|gemini-2.5-pro (reasoning mode)|claude-sonnet-4-5-20250929 (reasoning mode)|Average|
> |---|---|---|---|---|---|---|---|---|---|
> |SimpleQA|0.648|0.689|0.974|0.853|0.125|0.119|0.521|0.823|0.594|
> |Pure gambling|0.780|0.990|0.340|0.990|0.760|0.760|1.000|1.000|0.828|
> |Ideal refusal proportions|1.000|1.000|1.000|1.000|1.000|1.000|1.000|1.000|1.000|
>
> There are mainly two findings:
> 1. LMs over-refuse in the $(1,0,0)$ setting, and over-answer in the $(0,-1,0)$ setting.
> 2. LMs’ over-refusing and over-answering behaviors are mostly mitigated in the pure gambling setting, showing that they mostly apply the answer-or-refuse skill better when presented with a “pure” decision problem. Some exceptions include Gemini models in $(1,0,0)$ and claude-3-5-haiku in $(0,-1,0)$.
>
> **These findings align with our original findings in the multiple-choice setting, suggesting that the performance gap we found is not an artifact of four-option multiple-choice QA.**
>
> ***(b) Measuring LMs’ abilities to maximize expected reward (analogue to Table 1):***
> |Method / Model|gpt-4o-mini-2024-07-18|gpt-4o-2024-08-06|claude-3-5-haiku-20241022|claude-sonnet-4-5-20250929 (no reasoning)|gemini-2.5-flash (no reasoning)|gemini-2.5-flash (reasoning mode)|gemini-2.5-pro (reasoning mode)|claude-sonnet-4-5-20250929 (reasoning mode)|Average|
> |---|---|---|---|---|---|---|---|---|---|
> |No-risk prompt|-6.637|-4.333|-1.551|-3.46|-5.085|-5.165|-2.979|-3.331|-4.067625|
> |Risk-informing|-4.239|-0.95|-0.11|-0.184|-4.607|-4.336|-2.638|-0.292|-2.1695|
> |Stepwise prompt|-5.395|-1.843|-0.811|-2.289|-4.793|-4.715|-2.924|-2.206|-3.122|
> |Prompt chaining|-1.314|-0.443|-0.317|-0.212|-4.365|-4.241|-2.634|-0.079|-1.700625|
>
> The results again align closely with the multiple-choice findings, with prompt chaining mostly performing better, showing that most LMs require skill decomposition to fully leverage their individual skills.
>
> **References:**
>
> [1] Wei, Jason, et al. "Measuring short-form factuality in large language models." arXiv preprint arXiv:2411.04368 (2024).
>
> [2] Haas, Lukas, et al. "SimpleQA Verified: A reliable factuality benchmark to measure parametric knowledge." arXiv preprint arXiv:2509.07968 (2025).

---

> ### Author Response · Authors · 2025-11-24
> **Rebuttal to Weakness 2 (W2): Regarding the Prompt Chaining Method**
>
> > The proposed "prompt chaining" method also lacks generalizability. The paper makes an insightful observation about the "knowing-doing gap" in an LLM's application of expected-value reasoning. However, the proposed solution feels very specific and applies only to this particular, quantitative risk structure for multiple-choice tasks. To improve the impact of this work, I suggest authors explore more generalizable (e.g. to more variety of risk-aware decision making problems) prompting or post-training methods to bridge this kind of knowing-doing gap during decision making
>
> First, thank you for your acknowledgement that the "knowing-doing gap" is an insightful observation. We really appreciate your precise understanding of what we're trying to reveal.
>
> Also, thank you for raising the concern about the generalizability of "prompt chaining". **We'd like to emphasize that we do not suggest prompt chaining as an ultimate solution to solve the "knowing-doing gap"**. The performance gain is best considered as a by-product of our new understanding that current LMs still require human guidance to perform related skills effectively. We will emphasize this perspective more explicitly in the paper.
>
> We totally agree that certain post-training techniques (e.g., supervised fine-tuning or RL) would be possible solutions to mitigate the issue. At the same time, we remain firm in our stance that our three main contributions (line 84 - 96) are already sufficiently self-contained as a research work.

---

> ### Author Response · Authors · 2025-11-25
> **Rebuttal to Question 1 (Q1): Availability of Code**
>
> > (Q1) The paper mentioned “The code is attached as a .zip file in the submission”, but does not seem to be available on OpenReview. Could the authors please provide the code?
>
> Thank you for pointing out this issue. We have uploaded our code and README as a `.zip` file to the supplementary material. Feel free to let us know if you encounter any issues with the code!

---

> ### Author Response · Authors · 2025-11-25
> **Rebuttal to Question 2 (Q2): Relationship between Calibration and Final Scores**
>
> > Appendix H shows that better confidence estimation (calibration) improves final scores. This conclusion is based on an experiment comparing two specific prompt variants for the confidence-estimation step. I'm curious if authors have experimented with any other prompt variations for this step? I am wondering how robust this conclusion is and whether it holds across different ways of eliciting confidence
>
> We appreciate your thorough reading of our paper and for raising this nuanced question. When we were doing the original work, we only experimented with the two prompt variants mentioned in the Appendix. We agree that the robustness of this conclusion requires further experiments and discussions, which we provide as follows:
>
> **As you mentioned, we wonder whether this conclusion holds across different ways of eliciting confidence**. Therefore, instead of using “verbalized confidence” [1] as in our original experiments, we tried the “P(True)” [2] methodology. Specifically, we instruct LMs with the following prompt in the experiments on SimpleQA verified:
>
> ```
> Here is a question and your predicted answer.
>
> Question: {question}
> Your predicted answer: {predicted_answer}
>
> Is your predicted answer correct? Please directly output a single token YES or NO.
> ```
>
> Then, we take the token log probabilities of the “YES” and “NO” tokens, and compute the softmax probability of the “YES” token as the elicited confidence. We list the difference of the (1) calibration performance and (2) final scores between “verbalized confidence” and “P(True)” as follows:
>
> ***Note:*** Some LMs do not provide access to token log probabilities (e.g., Claude models), so we only ran experiments on the LMs that provide this access.
>
> |$(r_{cor},r_{inc},r_{ref})=(1,-8,0)$ Performance Difference between Methods / Models|gpt-4o-mini-2024-07-18|gpt-4o-2024-08-06|gemini-2.5-flash (no reasoning mode)|gemini-2.5-flash (reasoning mode)|gemini-2.5-pro (reasoning mode)|
> |---|---|---|---|---|---|
> |(1) ECE (↓): P(True) - Verbalized Confidence|**-0.3771**|-0.0654|**-0.0846**|**-0.0001**|0.0028|
> |(2) Final scores (↑): P(True) - Verbalized Confidence|**0.5850**|-1.4850|**0.3550**|**0.0650**|0.0490|
>
> For (1) calibration performance, we use Expected Calibration Error (ECE) [3] as the metric, which we also used in Appendix H. Lower ECE means better calibration, so the negative ECE values in the table denote the cases where “P(True)” performs better than “verbalized confidence”. One can see that in **3 out of 5** cases, better calibration leads to higher final scores. In Appendix H, better calibration leads to higher final scores in **10 out of 12** cases.
>
> While these results show that better calibration (mostly) improves final scores, it is not always the case. Therefore, we conduct deeper analysis to discuss why. We analyze the case of `gpt-4o-2024-08-06` in the table above, since it is a case where the final score degrades a lot despite P(True)’s better calibration.
>
> First of all, since $(r_{cor},r_{inc},r_{ref})=(1,-8,0)$, we can compute the answer-or-refuse confidence threshold ($p$) as $p=\frac{-r_{inc}}{r_{cor}-r_{inc}}=\frac{8}{9}$. Therefore, a perfect case for the LM is that it assign $p>\frac{8}{9}$ to all correct instances and assign $p<\frac{8}{9}$ to all incorrect instances, such that it gets all rewards and avoid all penalties. We now list how “P(True)” and “verbalized confidence” compare to this perfect case as the following matrices:
> |Perfect case|p > 8/9 (answer)|p < 8/9 (refuse)|
> |---|---|---|
> |Correct|339|0|
> |Incorrect|0|661|
>
> |Verbalized confidence|p > 8/9|p < 8/9|
> |---|---|---|
> |Correct|42|297|
> |Incorrect|60|601|
>
> |P(True)|p > 8/9|p < 8/9|
> |---|---|---|
> |Correct|192|147|
> |Incorrect|266|395|
>
> Although P(True) (avg. confidence = 63.53%) achieves better calibration than verbalized confidence (avg. confidence = 80.80%), the method P(True) assigns relatively more incorrect instances to $p>\frac{8}{9}$, causing this method to lose more final scores. Therefore, our current conclusion is that the better calibration generally aligns with better final scores, but the exact result depends on how the calibration metric (e.g., ECE) actually aligns with the matrix of "correctness vs. $p\lessgtr\frac{-r_{inc}}{r_{cor}-r_{inc}}$".
>
> **References:**
>
> [1] Tian, Katherine, Eric Mitchell, Allan Zhou, Archit Sharma, Rafael Rafailov, Huaxiu Yao, Chelsea Finn, and Christopher D. Manning. "Just Ask for Calibration: Strategies for Eliciting Calibrated Confidence Scores from Language Models Fine-Tuned with Human Feedback." In The 2023 Conference on Empirical Methods in Natural Language Processing.
>
> [2] Kadavath, Saurav, Tom Conerly, Amanda Askell, Tom Henighan, Dawn Drain, Ethan Perez, Nicholas Schiefer et al. "Language models (mostly) know what they know." arXiv preprint arXiv:2207.05221.
>
> [3] Guo, Chuan, Geoff Pleiss, Yu Sun, and Kilian Q. Weinberger. "On calibration of modern neural networks." In International conference on machine learning, pp. 1321-1330.

---

> ### Comment · Reviewer_m6Nb · 2025-11-27
>
> ​Thank you for the thoughtful response and for addressing my questions. I appreciate the effort put into the additional experiments, particularly the inclusion of SimpleQA and the further analysis on prompt sensitivity.
>
> ​I acknowledge your core finding that LLMs struggle to perform simple decision-making tasks given a specified risk structure, despite possessing the individual skills to do so. This is an interesting observation and exposures an important gap in llm's decision making capability.
>
> ​However, I maintain my concern regarding the risk structure being a bit artificial, and contribution of the finding of decision making gap under this structure being somewhat limited. It would be interesting to see either extending the risk structure into more context dependent scenarios, or dive deeper into more general form of knowing doing gap in decision making.
>
> I have decided to keep my current score.

---

### Official Review · Reviewer_5kFo · 2025-10-30

**Soundness:** 2
**Presentation:** 3
**Contribution:** 2
**Rating:** 4
**Confidence:** 3

**Summary:**

This paper introduces a framework to evaluate language models (LMs) in risk-aware decision-making scenarios, where models can choose to answer or refuse based on a specified reward structure (correct, incorrect, refusal). Experiments across multiple datasets and LMs show that models often over-answer in high-risk settings and over-refuse in low-risk ones. The authors propose a skill-decomposition method (via prompt chaining) that guides LMs through problem solving, confidence estimation, and expected-value reasoning, improving outcomes.

**Strengths:**

* The paper tackles a critical aspect of LM reliability which is relevant for real-world deployment.

* The reward structure is formalized clearly and enables systematic testing of LM decision behavior.

* The empirical observation that LMs can perform expected-value reasoning in isolation but fail to apply it in mixed settings is important.

* Prompt chaining is a simple, effective strategy to improve reward maximization under risk. Results show meaningful gains in high-risk settings.

**Weaknesses:**

1. All tasks are four-option multiple-choice questions. The framework’s applicability to open-ended, real-world tasks is untested.

2. The refusal reward is always set to 0. In practice, refusal often carries non-zero cost (latency/missed opportunity), which may alter optimal strategies.

3. No statistical significance reporting: Results lack variance or confidence intervals, which reduces interpretability of small differences across methods.

4. Incremental novelty: The main techniques of prompt decomposition and risk prompting are adaptations of known strategies. The paper does not introduce new modeling or training methods.

**Questions:**

1. How would results change if refusal incurred a cost (e.g., r_ref = –1)? Would LMs still over-defer in low-risk settings?

2. Have you compared your decomposition method to simpler alternatives like using softmax confidence thresholds or calibrated logits?

3. How sensitive is performance to the wording of prompts or to different chain-of-thought styles? Could improved prompting alone close the gap?

---

> ### Author Response · Authors · 2025-11-24
> **Rebuttal to Weakness 1 (W1)**
>
> > (W1) All tasks are four-option multiple-choice questions. The framework’s applicability to open-ended, real-world tasks is untested.
>
> We address your concern by the following two points: **(1) Why we use multi-choice questions** and **(2) How the findings of our framework apply to open-ended tasks**
>
> **(1) Why we use four-option multiple-choice tasks in the core analysis**
>
> As discussed in the paper (lines 137–145), the multiple-choice setting provides a mathematically useful structure: with the number of options (e.g., $K = 4$), we can compute the expected reward from random guessing: $r_{guess} = \frac{1}{K}r_{cor} + \frac{K-1}{K}r_{inc}$
>
> Knowing $r_{guess}$ allows us to derive the ideal refusal policy under certain risk structures. For example, when $r_{\text{guess}} > r_{\text{ref}}$ (defined as low-risk settings in our paper), an ideal agent should always answer. This characterization lets us quantify the gap between the model’s empirical refusal behavior and the optimal behavior shown in Figure 3 (lines 230–236). Therefore, **we intentionally use multiple-choice questions in our analysis, because the nice mathematical property allows comparisons of model behaviors against the ideal**.
>
> **(2) New evidence from free-form generation tasks**
>
> While free-form generation tasks do not offer the same mathematical property, we agree that testing them makes our work more robust. To address this, we conducted new experiments on SimpleQA Verified [1,2], which is a free-form QA task that measures LMs’ refusal ability to avoid hallucination when uncertain. We ran the following two main experiments:
>
> ***(a) Measuring refusal behavior in free-form QA vs. pure gambling (analogue to Figure 3)***
>
> We re-ran the same experiments in Figure 3 with two risk structures $(r_{cor}, r_{inc}, r_{ref}) = (1,0,0)$ and $(0,-1,0)$. The ideal refusal proportion for $(1,0,0)$ is 0% since the expected reward of answering is always larger than refusal. Likewise, the ideal refusal proportion for $(0,-1,0)$ is 100%. The results are as follows:
>
> |(r_cor, r_inc, r_ref) = (1, 0, 0)|gpt-4o-mini-2024-07-18|gpt-4o-2024-08-06|claude-3-5-haiku-20241022|claude-sonnet-4-5-20250929 (no reasoning)|gemini-2.5-flash (no reasoning)|gemini-2.5-flash (reasoning mode)|gemini-2.5-pro (reasoning mode)|claude-sonnet-4-5-20250929 (reasoning mode)|Average|
> |---|---|---|---|---|---|---|---|---|---|
> |SimpleQA|0.298|0.538|0.952|0.583|0.088|0.060|0.013|0.484|0.377|
> |Pure gambling|0.020|0.000|0.000|0.000|0.160|0.130|0.020|0.000|0.041|
> |Ideal refusal proportions|0.000|0.000|0.000|0.000|0.000|0.000|0.000|0.000|0.000|
>
> |(r_cor, r_inc, r_ref) = (0, -1, 0)|gpt-4o-mini-2024-07-18|gpt-4o-2024-08-06|claude-3-5-haiku-20241022|claude-sonnet-4-5-20250929 (no reasoning)|gemini-2.5-flash (no reasoning)|gemini-2.5-flash (reasoning mode)|gemini-2.5-pro (reasoning mode)|claude-sonnet-4-5-20250929 (reasoning mode)|Average|
> |---|---|---|---|---|---|---|---|---|---|
> |SimpleQA|0.648|0.689|0.974|0.853|0.125|0.119|0.521|0.823|0.594|
> |Pure gambling|0.780|0.990|0.340|0.990|0.760|0.760|1.000|1.000|0.828|
> |Ideal refusal proportions|1.000|1.000|1.000|1.000|1.000|1.000|1.000|1.000|1.000|
>
> There are mainly two findings:
> 1. LMs over-refuse in the $(1,0,0)$ setting, and over-answer in the $(0,-1,0)$ setting.
> 2. LMs’ over-refusing and over-answering behaviors are mostly mitigated in the pure gambling setting, showing that they mostly apply the answer-or-refuse skill better when presented with a “pure” decision problem. Some exceptions include Gemini models in $(1,0,0)$ and claude-3-5-haiku in $(0,-1,0)$.
>
> **These findings align with our original findings in the multiple-choice setting, suggesting that the performance gap we found is not an artifact of four-option multiple-choice QA.**
>
> ***(b) Measuring LMs’ abilities to maximize expected reward (analogue to Table 1):***
> |Method / Model|gpt-4o-mini-2024-07-18|gpt-4o-2024-08-06|claude-3-5-haiku-20241022|claude-sonnet-4-5-20250929 (no reasoning)|gemini-2.5-flash (no reasoning)|gemini-2.5-flash (reasoning mode)|gemini-2.5-pro (reasoning mode)|claude-sonnet-4-5-20250929 (reasoning mode)|Average|
> |---|---|---|---|---|---|---|---|---|---|
> |No-risk prompt|-6.637|-4.333|-1.551|-3.46|-5.085|-5.165|-2.979|-3.331|-4.067625|
> |Risk-informing|-4.239|-0.95|-0.11|-0.184|-4.607|-4.336|-2.638|-0.292|-2.1695|
> |Stepwise prompt|-5.395|-1.843|-0.811|-2.289|-4.793|-4.715|-2.924|-2.206|-3.122|
> |Prompt chaining|-1.314|-0.443|-0.317|-0.212|-4.365|-4.241|-2.634|-0.079|-1.700625|
>
> The results again align closely with the multiple-choice findings, with prompt chaining mostly performing better, showing that most LMs require skill decomposition to fully leverage their individual skills.
>
> **References:**
>
> [1] Wei, Jason, et al. "Measuring short-form factuality in large language models." arXiv preprint arXiv:2411.04368 (2024).
>
> [2] Haas, Lukas, et al. "SimpleQA Verified: A reliable factuality benchmark to measure parametric knowledge." arXiv preprint arXiv:2509.07968 (2025).

---

> ### Author Response · Authors · 2025-11-25
> **Rebuttal to Weakness 2 (W2) and Question 1 (Q1)**
>
> > (W2) The refusal reward is always set to 0. In practice, refusal often carries non-zero cost (latency/missed opportunity), which may alter optimal strategies.
>
> > (Q1) How would results change if refusal incurred a cost (e.g., r_ref = –1)? Would LMs still over-defer in low-risk settings?
>
> Thank you for the question. We re-ran experiments in Figure 3 with $r_{ref} = -1$ in low-risk settings (where the ideal refusal proportion is 0 since $r_{guess} > r_{ref}$) and the results are as follows:
>
> |(r_cor, r_inc, r_ref) = (4, -1, -1)|gpt-4o-mini-2024-07-18|gpt-4o-2024-08-06|claude-3-5-haiku-20241022|claude-sonnet-4-5-20250929 (no thinking)|gemini-2.5-flash (no thinking)|gemini-2.5-flash (reasoning mode)|gemini-2.5-pro (reasoning mode)|claude-sonnet-4-5-20250929 (reasoning mode)|Average|
> |---|---|---|---|---|---|---|---|---|---|
> |GPQA|0.027|0.009|0.029|0.000|0.143|0.045|0.004|0.000|0.032|
> |Pure gambling|0.060|0.000|0.000|0.000|0.090|0.060|0.000|0.000|0.026|
> |Ideal refusal proportion|0.000|0.000|0.000|0.000|0.000|0.000|0.000|0.000|0.000|
>
> |(r_cor, r_inc, r_ref) = (8, -1, -1)|gpt-4o-mini-2024-07-18|gpt-4o-2024-08-06|claude-3-5-haiku-20241022|claude-sonnet-4-5-20250929 (no thinking)|gemini-2.5-flash (no thinking)|gemini-2.5-flash (reasoning mode)|gemini-2.5-pro (reasoning mode)|claude-sonnet-4-5-20250929 (reasoning mode)|Average|
> |---|---|---|---|---|---|---|---|---|---|
> |GPQA|0.018|0.013|0.018|0.002|0.123|0.045|0.000|0.000|0.027|
> |Pure gambling|0.020|0.000|0.010|0.000|0.090|0.040|0.000|0.000|0.020|
> |Ideal refusal proportion|0.000|0.000|0.000|0.000|0.000|0.000|0.000|0.000|0.000|
>
> |(r_cor, r_inc, r_ref) = (1, 0, -1)|gpt-4o-mini-2024-07-18|gpt-4o-2024-08-06|claude-3-5-haiku-20241022|claude-sonnet-4-5-20250929 (no thinking)|gemini-2.5-flash (no thinking)|gemini-2.5-flash (reasoning mode)|gemini-2.5-pro (reasoning mode)|claude-sonnet-4-5-20250929 (reasoning mode)|Average|
> |---|---|---|---|---|---|---|---|---|---|
> |GPQA|0.025|0.011|0.022|0.000|0.112|0.047|0.000|0.000|0.027|
> |Pure gambling|0.020|0.000|0.010|0.000|0.080|0.080|0.000|0.010|0.025|
> |Ideal refusal proportion|0.000|0.000|0.000|0.000|0.000|0.000|0.000|0.000|0.000|
>
> The results show that LMs indeed refuse less than when $r_{ref}=0$, but they still over-refuse in these settings. Moreover, most LMs’ refusal proportion is closer to the ideal refusal proportion in the pure gambling setting, showing the “knowing-doing gap” of expected value reasoning still exists.

---

> ### Author Response · Authors · 2025-11-25
> **Rebuttal to Weakness 3 (W3): No Reporting of Variance**
>
> > (W3) No statistical significance reporting: Results lack variance or confidence intervals, which reduces interpretability of small differences across methods.
>
> We want to clarify that we actually reported performance variance by standard deviations of our multi-prompt evaluation results in the Appendix (Table 10). We would integrate these numbers into Table 1 and 2 into the main text.

---

> ### Author Response · Authors · 2025-11-25
> **Rebuttal to Weakness 4 (W4)**
>
> > (W4) Incremental novelty: The main techniques of prompt decomposition and risk prompting are adaptations of known strategies. The paper does not introduce new modeling or training methods.
>
> Our main contributions are that we are the first to (1) design the evaluation framework to investigate whether LMs can adjust their answer-or-defer policy when informed of different risk structures, and (2) reveal their failure modes of “over-answering” or “over-deferring”. **We believe that our novelty mainly lies in this topic and the empirical findings that highlight these previously under-examined issues.**
>
> Regarding the method of skill decomposition via prompt chaining, it mainly serves as a way to show that LMs could indeed perform these individual skills, yet struggle to combine these skills effectively by themselves. **We consider this finding itself as a valuable contribution, and we do not claim methodological novelty for the decomposition technique.**

---

> ### Author Response · Authors · 2025-11-25
> **Rebuttal to Question 3 (Q3)**
>
> > (Q3) How sensitive is performance to the wording of prompts or to different chain-of-thought styles? Could improved prompting alone close the gap?
>
> As noted in lines 412 - 416 of our paper, **we conducted a multi-prompt evaluation to ensure that our results are robust to variations in prompt wording**. The performance of individual prompt variants, along with their standard deviations, is reported in Tables 7–10. Across all prompt styles, we consistently observe a performance gap between prompt chaining and the risk-informing prompt in the high-risk settings. **The persistence of this gap suggests that the issue is not merely prompt sensitivity, but a deeper difficulty in integrating the relevant skills.**

---

> ### Author Response · Authors · 2025-11-26
> **Rebuttal to Question 2 (Q2): Compare Decomposition to Simpler Alternatives (Token Softmax)**
>
> Thank you for this suggestion. We agree that using softmax probabilities over the four option tokens (i.e., A, B, C, and D) to determine answer-or-defer decisions is a reasonable method to run. We re-run experiments in Table 1 and 2 and provide the results as follows (with the upper left cell being $Dataset$; $(r_{cor},r_{inc},r_{ref})$):
>
> Note:
> 1. The probability threshold ($p$) for deciding whether to answer or defer is calculated as follows:
> $p \cdot r_{cor} + (1 - p) \cdot r_{inc} = r_{ref} \Leftrightarrow p = \frac{r_{ref} - r_{inc}}{r_{cor} - r_{inc}}$
> For example, $p = 0.8$ when the risk structure $(r_{cor},r_{inc},r_{ref}) = (1,-4,0)$, so answering requires softmax probability of the predicted option token to be at least 80%.
>
> 2. Note that a limitation of this method is the access to the token log probabilities, which might not always be available for all close-weight models. Therefore, we ran experiments with `gpt-4o-mini` and `gpt-4o` as representative LMs, since they (1) are models used in our original experiments, (2) are still served through API and not deprecated, and (3) provide access to token log probabilities. If the reviewers want to see results with other LMs, we can run experiments on those.
>
> 3. The first four rows are from results in our original paper, while the last row is the new results we obtain.
>
> |GPQA; (1,-8,0)|gpt-4o-mini-2024-07-18|gpt-4o-2024-08-06|
> |---|---|---|
> |No-risk prompt|-4.391|-3.336|
> |Risk-informing|-4.273|-3.210|
> |Stepwise prompt|-4.334|-3.294|
> |Prompt chaining|-0.969|-0.054|
> |Softmax confidence threshold|-3.935|-2.757|
>
> |GPQA; (1,-4,0)|gpt-4o-mini-2024-07-18|gpt-4o-2024-08-06|
> |---|---|---|
> |No-risk prompt|-1.998|-1.416|
> |Risk-informing|-1.938|-1.412|
> |Stepwise prompt|-1.972|-1.552|
> |Prompt chaining|-0.435|-1.115|
> |Softmax confidence threshold|-1.788|-1.152|
>
> |MedQA; (1,-8,0)|gpt-4o-mini-2024-07-18|gpt-4o-2024-08-06|
> |---|---|---|
> |No-risk prompt|-0.818|0.147|
> |Risk-informing|-0.789|0.159|
> |Stepwise prompt|-0.835|0.166|
> |Prompt chaining|-0.083|0.475|
> |Softmax confidence threshold|-0.806|0.150|
>
> |MedQA; (1,-4,0)|gpt-4o-mini-2024-07-18|gpt-4o-2024-08-06|
> |---|---|---|
> |No-risk prompt|-0.010|0.526|
> |Risk-informing|0.014|0.537|
> |Stepwise prompt|-0.173|0.531|
> |Prompt chaining|0.267|0.565|
> |Softmax confidence threshold|-0.011|0.524|
>
> |MMLU; (1,-8,0)|gpt-4o-mini-2024-07-18|gpt-4o-2024-08-06|
> |---|---|---|
> |No-risk prompt|-0.668|-0.115|
> |Risk-informing|-0.651|-0.076|
> |Stepwise prompt|-0.683|-0.169|
> |Prompt chaining|-0.016|0.327|
> |Softmax confidence threshold|-0.645|-0.091|
>
> |MMLU; (1,-4,0)|gpt-4o-mini-2024-07-18|gpt-4o-2024-08-06|
> |---|---|---|
> |No-risk prompt|0.073|0.380|
> |Risk-informing|0.084|0.367|
> |Stepwise prompt|0.046|0.317|
> |Prompt chaining|0.295|0.444|
> |Softmax confidence threshold|0.079|0.357|
>
> Discussion:
> 1. By comparing rows of “softmax confidence threshold” to “risk-informing”, one can see that the softmax method performs better on GPQA, while the “risk-informing” method (mostly) performs better on MedQA and MMLU. Therefore, **using “softmax confidence threshold” does not always provide improvements on top of just informing LMs about the risk in the instructions.**
>
> 2. By comparing rows of “softmax confidence threshold” to “prompt chaining”, one can see that “prompt chaining” always performs better. It suggests that decomposing the skills (via prompt chaining) for the LMs generally provides further advantages than other methods, including alternatives such as "using softmax token probabilities and compare them to thresholds". **This reinforces our original findings.**
>
> We hope that these new results address your question.

---

### Official Review · Reviewer_ZUJw · 2025-11-01

**Soundness:** 3
**Presentation:** 3
**Contribution:** 2
**Rating:** 4
**Confidence:** 3

**Summary:**

This paper investigates the problem whether LLMs can make risk-aware decision based on a given risk structure (rewards for correct answers, penalties for errors, and payoff for refusal). The authors introduce an evaluation framework that systematically varies risk settings while keeping tasks fixed, allowing them to isolate LLMs’ decision-making policies from their raw task-solving ability. Several interesting findings are given from an evaluation of several LLM families.

**Strengths:**

This paper aims to address a critical problem in LLM application: how to make LLMs risk-aware in decision making. It proposes an evaluation framework to investigate the impact of varied risk structures for constant tasks. The study validates the effectiveness of this approach on multiple-choice question tasks, finding that LLMs tend to over-answer in high-risk scenarios while demonstrating excessive deferral in low-risk environments. A skill decomposition-based prompt optimization strategy is proposed to address these issues, which is simple, effective, and does not require model retraining.

**Weaknesses:**

**Limited Task**: The study is restricted to multiple-choice questions. It remains unclear whether the findings generalize to open-ended generation or real-world interactive tasks.

**Simplified Risk Structure**: The risk settings are static and pre-defined. Real-world risk can be dynamic and context-dependent. It is unclear whether the method generalize to more complex settings.

**Questions:**

**Generalization beyond multiple-choice**: How would you expect the results to transfer to free-form generation or real-world decision-making tasks?

**Dynamic risk adaptation**: Have the authors considered scenarios where the risk structure changes during interaction? Can LMs adapt to dynamic risk settings? Despite the settings given in the paper, further more complex risk structures are welcome for validation.

**Human-in-the-Loop scenarios**: In high-stakes applications, should LMs always defer to humans when uncertain? How does the framework incorporate human oversight?

**Long-term decision behavior**:  The work mainly focuses on single-step decision task. How might these findings extend to multi-step planning agents scenarios that must reason about risk over a sequence of actions?

---

> ### Author Response · Authors · 2025-11-21
> **Rebuttal to Weakness 1 (W1) and Question 1 (Q1)**
>
> > (W1) Limited Task: ... It remains unclear whether the findings generalize to open-ended generation ...
>
> > (Q1) Generalization beyond multiple-choice: ...
>
> We would like to address your concern with the following two points:
>
> **(1) Reasons behind using multiple-choice questions:** As mentioned in our paper (line 137 - 145), this setting enables us to compute the expected reward from random guessing:
>
> $r_{guess} = \frac{1}{K}r_{cor} + \frac{K-1}{K}r_{inc}$
>
> where $K = 4$ is the number of options. With the value of $r_{guess}$ determined, one can naturally see that when $r_{guess} > r_{ref}$ (random guessing yields higher expected reward than refusal), the ideal policy is to always answer. This property enables us to study the gap between “LMs’ refusal proportions” and the “ideal refusal proportions” shown in Figure 3 (line 230 - 236).
>
> **(2) New experiment results on free-form generation:** Although free-form generation tasks do not have the nice property mentioned above, we agree that experiment results on these can strengthen our findings. Therefore, we conducted additional experiments on SimpleQA verified [1,2], which is a free-form question answering (QA) task that is sufficiently difficult for frontier LMs. It also measures LMs’ refusal to evaluate their ability to avoid hallucination when uncertain, which is aligned with our setting. We ran the following two main experiments in our original paper:
>
> ***(a) Measuring LMs’ refusal proportions from SimpleQA vs. pure gambling (Figure 3):***
> We re-ran the same experiments in Figure 3 on SimpleQA verified [1,2] with two risk structures $(r_{cor}, r_{inc}, r_{ref}) = (1,0,0)$ and $(0,-1,0)$. The ideal refusal proportion for $(1,0,0)$ is 0% since the expected reward of answering is always larger than refusal. Likewise, the ideal refusal proportion for $(0,-1,0)$ is 100%. The results are as follows:
>
> |(r_cor, r_inc, r_ref) = (1, 0, 0)|gpt-4o-mini-2024-07-18|gpt-4o-2024-08-06|claude-3-5-haiku-20241022|claude-sonnet-4-5-20250929 (no reasoning mode)|gemini-2.5-flash (no reasoning mode)|gemini-2.5-flash (reasoning mode)|gemini-2.5-pro (reasoning mode)|claude-sonnet-4-5-20250929 (reasoning mode)|Average|
> |---|---|---|---|---|---|---|---|---|---|
> |SimpleQA|0.298|0.538|0.952|0.583|0.088|0.060|0.013|0.484|0.377|
> |Pure gambling|0.020|0.000|0.000|0.000|0.160|0.130|0.020|0.000|0.041|
> |Ideal refusal proportions|0.000|0.000|0.000|0.000|0.000|0.000|0.000|0.000|0.000|
>
> |(r_cor, r_inc, r_ref) = (0, -1, 0)|gpt-4o-mini-2024-07-18|gpt-4o-2024-08-06|claude-3-5-haiku-20241022|claude-sonnet-4-5-20250929 (no thinking)|gemini-2.5-flash (no thinking)|gemini-2.5-flash (reasoning_effort='medium')|gemini-2.5-pro (reasoning_effort='medium')|claude-sonnet-4-5-20250929 (thinking_budget=8192)|Average|
> |---|---|---|---|---|---|---|---|---|---|
> |SimpleQA|0.648|0.689|0.974|0.853|0.125|0.119|0.521|0.823|0.594|
> |Pure gambling|0.780|0.990|0.340|0.990|0.760|0.760|1.000|1.000|0.828|
> |Ideal refusal proportions|1.000|1.000|1.000|1.000|1.000|1.000|1.000|1.000|1.000|
>
> There are mainly two findings:
> 1. LMs over-refuse in the $(1,0,0)$ setting, and over-answer in the $(0,-1,0)$ setting.
> 2. LMs’ over-refusing and over-answering behaviors are mostly mitigated in the pure gambling setting, showing that they mostly apply the answer-or-refuse skill better when presented with a “pure” decision problem. Some exceptions include Gemini models in $(1,0,0)$ and claude-3-5-haiku in $(0,-1,0)$.
>
> These findings align with our original findings in Section 3 on multi-choice questions.
>
> ***(b) Measuring LMs’ abilities to maximize expected reward (Table 1):***
> We also re-ran experiments in Table 1 on SimpleQA verified [1,2]:
>
> |Method / Model|gpt-4o-mini-2024-07-18|gpt-4o-2024-08-06|claude-3-5-haiku-20241022|claude-sonnet-4-5-20250929 (no reasoning mode)|gemini-2.5-flash (no reasoning mode)|gemini-2.5-flash (reasoning mode)|gemini-2.5-pro (reasoning mode)|claude-sonnet-4-5-20250929 (reasoning mode)|Average|
> |---|---|---|---|---|---|---|---|---|---|
> |No-risk prompt|-6.637|-4.333|-1.551|-3.46|-5.085|-5.165|-2.979|-3.331|-4.067625|
> |Risk-informing|-4.239|-0.95|-0.11|-0.184|-4.607|-4.336|-2.638|-0.292|-2.1695|
> |Stepwise prompt|-5.395|-1.843|-0.811|-2.289|-4.793|-4.715|-2.924|-2.206|-3.122|
> |Prompt chaining|-1.314|-0.443|-0.317|-0.212|-4.365|-4.241|-2.634|-0.079|-1.700625|
>
> The findings mostly align with results on multi-choice questions, with prompt chaining generally performing better, showing that most LMs require skill decomposition to fully leverage their individual skills.
>
> *Note that since some older LMs are deprecated by the model providers, we use the latest models to test whether frontier models still have this performance gap.*
>
> **References:**
>
> [1] Wei, Jason, et al. "Measuring short-form factuality in large language models." arXiv preprint arXiv:2411.04368 (2024).
>
> [2] Haas, Lukas, et al. "SimpleQA Verified: A reliable factuality benchmark to measure parametric knowledge." arXiv preprint arXiv:2509.07968 (2025).

---

> ### Author Response · Authors · 2025-11-23
> **Rebuttal to Weakness 1 (W2) and Question 1 (Q2)**
>
> > (W2) Simplified Risk Structure: The risk settings are static and pre-defined. Real-world risk can be dynamic and context-dependent. It is unclear whether the method generalizes to more complex settings.
>
> Thank you for raising this point. **We intentionally design the risk settings to be static and explicitly specified** because real-world risk-aware decision making involves two distinct components:
>
> **(1) Risk assessment: determining the reward/penalty structure for the given scenario.**
>
> As we note in the paper (line 69-72) and as emphasized by Kalai et al. (2025) [1], risk structures are inherently task- and stakeholder-dependent. Without explicit specification, **it is impractical for LMs themselves to infer the “correct” reward/penalty tradeoffs from context alone**, as these vary widely across applications and user groups. We quote the related paragraphs as follows:
>
> > (Our paper, line 69-72) … because the rewards and penalties are task-dependent and stakeholder-dependent, the exact reward-penalty ratios, which we refer to as risk structure, should be specified by humans.
>
> > (Kalai et al., 2025, page 14, second paragraph) ... The ideal penalty might reflect likely real-world harms, but that is impractical as it is specific to the problem, the target application, and the user group. Without transparent specification within the instructions, it would be difficult to achieve consensus among language-model creators on the correct thresholds.
>
> As a concrete example, let’s say we are using LMs to predict medical diagnosis given a patient profile. Even if the patient profile is completely the same, the consequences of an incorrect medical diagnosis differ drastically in the following two cases: (a) when the application is an educational tool to help medical students learn versus (b) when the application is deployed in emergency departments to assist medical doctors. Therefore, defining the risk structure is humans’ (application developers or deployers) responsibility.
>
> **(2) Decision making given a specified risk structure—choosing whether to answer or defer so as to maximize expected reward.**
>
> Once the risk structure is provided, a desirable LM should adapt its answer-or-defer policy based on its own uncertainty because the uncertainty level depends on the LM’s own capability for the task. **Whether current LMs can reliably adjust their behavior under different risk settings is precisely the open research question our work aims to answer.**
>
> Our scope focuses on (2), not because dynamic or context-inferred risk is unimportant, but because **even this simplified yet fundamental case—can LMs adapt at all when the risk is clearly specified?—has not previously been evaluated**. Answering this question is a critical first step before studying more complex forms of risk.
>
> > (Q2) Dynamic risk adaptation: Have the authors considered scenarios where the risk structure changes during interaction? Can LMs adapt to dynamic risk settings? Despite the settings given in the paper, further more complex risk structures are welcome for validation.
>
> We agree that dynamic or context-dependent risk structures are valuable extensions. However, since the core question of LM adaptability under static, clearly defined risk has not been examined before, **we deliberately scope our work to this simplified yet foundational setting**. Our framework provides a controlled environment in which researchers could manipulate risk structures to measure adaptability, and we see dynamic-risk scenarios as a natural next step for future research building on our results.
>
> **Reference:**
>
> [1] Kalai, Adam Tauman, et al. "Why language models hallucinate." arXiv preprint arXiv:2509.04664 (2025).
>
> **We hope this clarification addresses your concerns, and we would be happy to discuss further extensions if helpful.**

---

> ### Author Response · Authors · 2025-11-23
> **Rebuttal to Question 3 (Q3)**
>
> > (Q3) Human-in-the-Loop scenarios: In high-stakes applications, should LMs always defer to humans when uncertain? How does the framework incorporate human oversight?
>
> Our work focuses on evaluating how well LMs adjust their answer-or-defer policies under different risk structures and on providing a quantitative framework to measure this ability. **The question of who an LM should defer to is completely flexible**, as it depends entirely on the application’s requirements and constraints.
>
> In high-stakes settings, developers may indeed choose to route all uncertain cases to humans. In other contexts, different escalation paths may be preferable. For example, when the cost or efficiency is an important consideration, application developers can use a weaker / cheaper LM as the default and defer to a stronger / more expensive LM only when needed, with human oversight as the final fallback. Our framework is orthogonal to these choices: it evaluates the LM’s ability to recognize uncertainty and defer appropriately under the specified risk structure, regardless of the specific entity (human or model) it defers to.
>
> In short, **our contribution is in assessing when a model should defer, not prescribing to whom it must defer**. This design decision allows the framework to support a wide range of human-in-the-loop configurations.

---

> ### Author Response · Authors · 2025-11-23
> **Rebuttal to Question 4 (Q4)**
>
> > (Q4) Long-term decision behavior: The work mainly focuses on single-step decision task. How might these findings extend to multi-step planning agents scenarios that must reason about risk over a sequence of actions?
>
> Thank you for bringing up this question. First of all, we interpret your question about “reasoning about risk over a sequence of actions” as referring to scenarios in which an agent must evaluate an action’s longer-term implications (i.e., how current decisions bring downstream outcomes and cumulative risk). This type of risk assessment requires the agent to forecast how risks propagate over multiple steps.
>
> **Such multi-step risk reasoning primarily concerns risk assessment**, which we view as conceptually distinct from the focus of our work. As discussed in our response to weakness 2 (W2: Simplified Risk Structure), our work is focused on investigating a different question: given a specified risk structure, can LMs adjust their answer-or-defer policy appropriately? **Multi-step risk assessment itself is therefore outside the scope of our contribution.**
>
> That said, our findings can still extend to multi-step planning settings where the application permits humans to specify the relevant risk structure for each step during the multi-step interactions. In such environments, the model’s responsibility remains the same: to adjust its decision-making policy in accordance with the given risk structure. The main limitation is that the risk specification has to come from an external source rather than being inferred autonomously by the model.
>
> In summary, the scope of our work does not contain multi-step risk assessment, but the proposed framework remains applicable to multi-step agents provided that the design / application scenario allows risk assessment to be done externally.

---

### Author Response · Authors · 2025-12-02
**Rebuttal Summary by Authors**

We highlight the key strengths and contributions of our work as follows:

1. **Identifying a critical reliability issue in LMs’ risk-aware decision-making.** Our study reveals a systematic failure pattern in which LMs over-answer in high-risk settings and over-refuse in low-risk settings (reviewers `ZUJw`, `5kFo`, and `2K4H`)
2. **Diagnosing the underlying cause through a carefully designed experiment.** We provide a thoughtful experimental framework that uncovers the “knowing–doing” gap driving these failure patterns (reviewer `m6Nb`)
3. **Clear formalization.** The reward structure and evaluation setup are clearly defined, enabling systematic testing of LMs’ decision-making behaviors (reviewers `2K4H` and `5kFo`)
4. **Clear presentation which provides insightful observations** (reviewers `m6Nb` and `2K4H`)

We also summarized reviewers’ major concerns and how we addressed them as follows:

1. **Generalization beyond multi-choice questions** (reviewers `ZUJw`, `5kFo`, `m6Nb`, and `2K4H`): We clarified the rationale for using multiple-choice questions and conducted new experiments on a free-form generation task (SimpleQA-Verified). **The SimpleQA results align closely with the multiple-choice findings, reinforcing the robustness and generality of our conclusions.**
2. **Formulation of risk structure** (reviewers `ZUJw`, and `m6Nb`): Reviewers asked how our risk formulation maps to real-world settings. We clarified that real-world decision-making involves two capabilities: *(1) risk assessment* and *(2) making the optimal decision given the risk assessment*. We elaborated on the reasons why (1) should be humans’ responsibility, so **the scope of our work is focused on (2) intentionally.**

Lastly, we have also resolved all remaining minor concerns through clarifications or additional experiments. We hope the AC will consider these updates in the final assessment.

---

### Meta-Review · Area_Chair_Xprm · 2026-01-05

**Summary:**

This submission studies risk-aware “answer vs. refuse/deferral” decision making for LMs used as agents. It introduces an evaluation framework where tasks are held fixed while a human-specified risk structure is systematically varied, and models are evaluated by expected reward maximization. Across multiple datasets/models, the authors report a consistent pattern: models over-answer in high-risk settings (should refuse more) and over-refuse in low-risk settings (should answer more). They diagnose a “knowing–doing gap”: models can do expected-value reasoning in isolation (a “pure gambling” setup), but fail to apply it when coupled with task solving. A simple skill-decomposition / prompt chaining procedure (solve → estimate confidence → EV decision) improves expected reward reliably, and rebuttal adds experiments showing similar phenomena on a free-form QA benchmark (SimpleQA Verified), plus additional ablations (non-zero refusal cost; softmax-threshold baseline; prompt-sensitivity/variance reporting; confidence elicitation variants).

**Reviewer Concerns:**

Too MCQ-focused: core experiments are 4-option multiple choice; unclear transfer to open-ended / real agent tasks.
Risk setup feels artificial: fixed numeric is far from contextual, dynamic real-world risk.
Refusal isn’t free
Stats/robustness: wanted variance / significance and prompt-sensitivity checks
Skepticism about novelty
Better baselines

**Reviewer Scores:**

ZUJw (4): likely to move slightly up, perhaps to 5.
5kFo (4): many critiques addressed, could be 5 as well.
m6Nb (4): explicitly kept score
2K4H (4): also kept score

Seems like more of an reject score overall.

---

### Decision · Program_Chairs · 2026-01-26

Reject